# Ischemic Rescue Potential of Conditioned Medium Derived from Skeletal Muscle Cells-Seeded Electrospun Fiber-Coated Human Amniotic Membrane Scaffolds

**DOI:** 10.3390/ijms252111697

**Published:** 2024-10-30

**Authors:** Hanis Nazihah Hasmad, Abid Nordin, Shiplu Roy Chowdhury, Nadiah Sulaiman, Yogeswaran Lokanathan

**Affiliations:** 1Department of Tissue Engineering and Regenerative Medicine, Faculty of Medicine, National University of Malaysia, Kuala Lumpur 56000, Malaysia; hanis@supergenic.com.my (H.N.H.); nadiahsulaiman@ukm.edu.my (N.S.); 2Graduate School of Medicine, KPJ Healthcare University, Nilai 71800, Negeri Sembilan, Malaysia; m.abid@kpju.edu.my; 3Centre for Commercialization of Regenerative Medicine, Toronto, ON M5G 1M1, Canada; shiplu56@gmail.com; 4Advance Bioactive Materials-Cells UKM Research Group, Universiti Kebangsaan Malaysia, Bangi 43600, Selangor, Malaysia

**Keywords:** amnion, electrospun fiber, skeletal muscle cells, paracrine signaling, ischemia

## Abstract

Revascularization procedures such as percutaneous coronary intervention (PCI) and coronary artery bypass grafting (CABG) are crucial to restore blood flow to the heart and are used in the treatment of myocardial infarction (MI). However, these techniques are known to cause myocardial reperfusion injury in the ischemic heart. The present study aims to mimic ischemia–reperfusion injury in vitro on primary human cardiomyocytes (HCMs) and use the established injury model to study the rescue mechanism of skeletal muscle cell (SkM)-seeded electrospun fiber-coated human amniotic membrane scaffold (EF–HAM) on injured cardiomyocytes through paracrine secretion. An in vitro ischemia–reperfusion injury model was established by exposing the HCM to 5 h of hypoxia, followed by a 6 h reoxygenation period. Six different conditioned media (CM) including three derived from SkM-seeded EF–HAMs were introduced to the injured cells to investigate the cardioprotective effect of the CM. Cell survival analysis, caspase-3 and XIAP expression profiling, mitochondrial membrane potential analysis, and measurement of reactive oxygen species (ROS) were conducted to evaluate the outcomes of the study. The results revealed a significant increase in the viability of HCM exposed to H/R injury by 77.2% (*p* < 0.01), 111.8% (*p* < 0.001), 68.7% (*p* < 0.05), and 69.5% (*p* < 0.05) when supplemented with HAM CM, EF–HAM 3 min CM, EF–HAM 5 min CM, and EF–HAM 7 min CM, respectively. Furthermore, CM derived from SkM-seeded EF–HAM scaffolds positively impacted hypoxia-/reoxygenation-induced changes in caspase-3 expression, mitochondrial membrane potential, and reactive oxygen species generation, but not in XIAP expression. These findings suggest that EF–HAM composite scaffolds can exert antiapoptotic and cardioregenerative effects on primary human cardiomyocytes through the paracrine mechanism.

## 1. Introduction

The optimal management of myocardial infarction (MI) remains a matter of ongoing debate within the scientific community. Existing therapeutic approaches predominantly address the consequences of MI, rather than focusing on regenerating lost myocardium or repairing damaged cardiomyocytes. Notably, revascularization techniques, such as percutaneous coronary intervention (PCI) and coronary artery bypass grafting (CABG), play a pivotal role in facilitating early revascularization to the infarcted myocardium, offering an opportunity for viable ischemic cells to recover while preventing the further enlargement of the ischemic myocardial zone [1,2].

Revascularization procedure s such as PCI and CABG crucial for restoring blood flow and oxygen delivery to ischemic tissues, are, however, associated with an unintended consequence known as myocardial ischemia–reperfusion injury (I/RI). This phenomenon occurs during the restoration of blood flow and oxygen delivery, which can exacerbate the inflammatory immune response activated during the ischemic phase. The inflammatory response is a significant contributor to the damage seen in I/RI. Besides that, the accumulation of intracellular sodium, hydrogen, and calcium ions at the ischemic site induces tissue acidosis, which can lead to cytotoxicity during reperfusion due to the rapid correction of acidosis [1]. Other key mechanisms contributing to the pathophysiology of I/RI include calcium overload, reactive oxygen species (ROS) burst, and mitochondrial dysfunction [3]. Consequently, there is a growing emphasis on understanding the molecular pathways involved in cardiomyocyte death to develop targeted interventions for myocardial regeneration and cardiomyocyte rescue within the infarcted heart [4].

Apoptosis, a highly regulated form of programmed cell death, emerges as a central focus in manipulating molecular pathways for therapeutic purposes in myocardial infarction (MI) and cardiac regeneration. The ability to modulate apoptosis, either preventatively or restoratively, presents an avenue for influencing the regeneration of lost myocardium or rescuing injured cardiomyocytes within the infarcted area. Specifically, hypoxia-inducible factor-1α (HIF-1α) serves as a biochemical marker indicating cardiomyocyte hypoxia, and its role becomes pivotal in understanding the pathophysiological mechanisms of I/RI. Increased oxidative stress, inflammatory responses, mitochondrial damage, and calcium overload during I/RI contribute to cell death, making in vitro induction of cardiomyocyte hypoxia a powerful tool for therapeutic screening of cardioprotective agents in the context of MI [5,6].

The executioner caspase, caspase-3, stands out as a major player in apoptosis, with its activation programming cell death through the proteolysis of DNA repair proteins, cytoskeletal proteins, and the inhibitor of caspase-activated DNase (ICAD) [6]. Imbalanced levels of caspase-3 indicate irreversible apoptosis [7], prompting many intervention studies to employ caspase-3 as a marker for assessing the effectiveness of interventions in inhibiting its expression. Similarly, the X-linked inhibitor of apoptosis (XIAP) emerges as a key member of the apoptosis inhibitor family, with demonstrated abilities to mitigate oxidative stress and inflammatory injury. The XIAP’s overexpression has been linked to ameliorating myocardial injury and apoptosis, acting as a negative regulator of cell apoptosis through interactions with caspase-3, caspase-7, and caspase-9. The activation of XIAP, achieved through the downregulation of specific miRNAs such as miR-134-5p and miR-181a-5p, serves to suppress oxidative stress and cardiomyocyte apoptosis induced by I/RI [8,9].

Mitochondrial damage stands as a critical pathophysiological mechanism in I/RI, leading to increased reactive oxygen species (ROS) production, rapid depolarization of the mitochondrial membrane potential, and impairment of ATP production in cells [10]. This study employs JC-1 staining as a marker for evaluating mitochondrial depolarization through the reduction of the red to green fluorescence intensity ratio [11]. Additionally, dihydroethidium (DHE), a molecule permeable to cells, is utilized as an oxidation marker. Under hypoxic conditions, the presence of O_2_^−^ oxidizes DHE to ethidium, staining the DNA-binding nuclei [12]. Consequently, elevated levels of DHE are commonly associated with myocardial injury and increased oxidative stress in MI [13].

In this context, the current study seeks to evaluate the rescue mechanism of primary human cardiomyocytes subjected to hypoxia–reoxygenation injury, facilitated by the paracrine secretion of skeletal muscle cell (SkM)-seeded tissue-engineered scaffolds made from electrospun fibers (EFs) and human amniotic membrane (HAM). In our previous study, the fabricated EF–HAM scaffolds demonstrated good biocompatibility and mechanical properties, making them suitable for cardiac tissue engineering applications [14,15]. In the subsequent study, SkM-seeded EF–HAM scaffolds exhibited potential angioinductive properties through a paracrine mechanism favorable for ischemic tissue repair and regeneration [16]. In this study, the induction and optimization of ischemia/reperfusion injury (I/RI) will be achieved under hypoxic conditions, with HIF-1α serving as an indicator of hypoxic injury. Four sets of apoptotic markers have been specifically chosen to represent the induction of the apoptosis cascade (caspase-3), inhibition of the apoptosis cascade (XIAP), mitochondrial membrane potential (JC-1), and oxidative stress (DHE). This comprehensive investigation aims to provide valuable insights into the potential of tissue-engineered EF–HAM scaffolds for mitigating the adverse effects of ischemia–reperfusion injury and enhancing the regeneration of infarcted myocardium following ischemic events.

## 2. Results

### 2.1. Effect of CM Derived from SkM-Seeded HAM and EF–HAM Scaffolds on the Viability and Proliferation of Healthy Primary Human Cardiomyocytes

The ability of CM derived from SkM-seeded HAM and EF–HAM scaffolds to maintain the viability and enhance the proliferation of primary human cardiomyocytes (HCM) was determined using CCK-8 Cell Viability Assay on the 2^nd^ and 5^th^ day of culture. The gross morphology of HCM under different CM treatments was monitored under the light microscope throughout the period of culture (Figure 1A). As shown in Figure 1B, there was no significant increase in the viability of HCM when supplemented with CM on the 2^nd^ day of culture as compared to the Non-CM control. Nevertheless, on the 5^th^ day of culture, HCM supplemented with EF–HAM 7 min CM showed significantly higher viability when compared to the Non-CM control (*p* < 0.05).

The proliferation of HCM treated with CM derived from SkM-seeded HAM and EF–HAM scaffolds was also assessed in this assay. According to Figure 1C, HCM exposed to the Non-CM control, Plain CM, and HAM CM had all experienced negative rates of cell proliferation due to a decline in the viability of the cells over the period of the cell culture. Nevertheless, HCM treated with different EF–HAM CM all showed positive rates of cell proliferation. Further analysis comparing all the groups showed that HCM supplemented with CM derived from all three different SkM-seeded EF–HAM scaffolds had a significantly higher proliferation rate compared to the Non-CM control (*p* < 0.05).

### 2.2. Establishment of In Vitro Cardiac Ischemia–Reperfusion Injury Model

The expression of HIF-1α, a specific marker for hypoxia, was studied to determine the minimum duration of hypoxia that should be exposed to HCM to achieve the highest expression of HIF-1α. Immunostained HCM showed the localization of HIF-1α to the cell nuclei when activated by hypoxia producing bright fluorescence signals (Figure 2A). As shown in Figure 2B, the expression of HIF-1α increased from 266.4 ± 14.5 at baseline 0 h hypoxia to 402.6 ± 31.5 after 3 h of hypoxia. The expression of HIF-1α peaked at the highest value of 504.9 ± 47.7 when HCM were exposed to hypoxia for 5 h. However, after 10 h of hypoxic treatment, HIF-1α expression in HCM declined to 359.3 ± 39.2. When HCM was further exposed to hypoxic conditions for 19 h, the expression of HIF-1α elevated to 412.0 ± 2.72 and remained high at 459.2 ± 42.5 after 24 h. Based on these results, it was apparent that a 5 h exposure to hypoxia was adequate in inducing the highest expression of HIF-1α in HCM.

The next step of optimization for the in vitro model of ischemia–reperfusion injury was to perform a cell toxicity assay on the post-hypoxic reoxygenated HCM. Following hypoxia, HCM was reoxygenated for a period up to 24 h before the cell viability was measured using CCK-8. Normoxic HCM culture was included in the assay as an experimental control and all the measurements were normalized to the control for comparative analysis. As Figure 2C shows, it is evident that the reoxygenation of HCM post-hypoxia significantly reduced the viability of cells in a time-dependent manner. Following reoxygenation, cell viability declined significantly from 100.0 ± 3.5% at normoxic to 54.1 ± 3.0% after 3 h, 46.3 ± 2.1% after 6 h, 39.6 ± 1.2% after 9 h, 27.8 ± 0.4% after 16 h, and 23.4 ± 1.8% after 24 h (*p* < 0.0001). Hence, a post-hypoxic reoxygenation time of 6 h was finally chosen as the optimal exposure time since it managed to reduce HCM viability by approximately 50%.

### 2.3. CM Derived from SkM-Seeded HAM and EF–HAM Scaffolds Enhanced Cell Survival and Rescued HCM Exposed to H/R Injury

The pro-survival effect of CM derived from SkM-seeded HAM and EF–HAM scaffolds on the viability of H/R-injured HCM was determined using a CCK-8 cell viability assay. As shown in Figure 3, the viability of H/R-injured HCM treated with Non-CM and Plain CM was significantly reduced by 58.4% (*p* < 0.001) and 42.5% (*p* < 0.01), respectively, when compared to the normoxic cells. As compared with the Non-CM control, supplementation of CM derived from SkM-seeded HAM and EF–HAM scaffolds significantly enhanced the viability of cells following H/R injury. Most importantly, the viability of HCM exposed to H/R injury increased significantly by 77.2% (*p* < 0.01), 111.8% (*p* < 0.001), 68.7% (*p* < 0.05), and 69.5% (*p* < 0.05) when supplemented with HAM CM, EF–HAM 3 min CM, EF–HAM 5 min CM, and EF–HAM 7 min CM, respectively, compared to the Plain CM-treated group.

### 2.4. CM Derived from SkM-Seeded EF–HAM 5 Min and 7 Min Scaffolds Ameliorated Apoptosis in HCM Following H/R Injury

Apoptosis is a highly regulated process controlled by the caspase family of proteins including caspase-3. To gain more insight into the antiapoptotic paracrine activity of SkM-seeded HAM and EF–HAM scaffolds, hypoxic HCMs were supplemented with CM derived from the engineered tissue scaffolds during the reoxygenation period, followed by the determination of caspase-3 expression using fluorescence microscopy (Figure 4). As demonstrated in Figure 4, HCM supplemented with the Non-CM control exhibited the highest expression of caspase-3 or number of apoptotic cells (60.5 ± 3.6%, *p* < 0.05) compared to those cultured under normoxic conditions (43.8 ± 1.3%). However, the most notable finding was the significant reduction of apoptosis in HCM treated with EF–HAM 5 min CM (38.8 ± 2.8%, *p* < 0.01) and EF–HAM 7 min CM (36.5 ± 4.7%, *p* < 0.01), where the number of apoptotic cells was significantly lower than those in HCM treated with Non-CM (60.5 ± 3.6%), Plain CM (54.9 ± 1.8%), and HAM CM (55.5 ± 2.9%). These results revealed the antiapoptotic potential of EF–HAM 5 min and EF–HAM 7 min scaffolds exerted on injured HCM via paracrine signaling.

### 2.5. Antiapoptotic Effect of EF–HAM CM on HCM with H/R Injury Was Not Mediated by XIAP

The expression of XIAP, an inhibitor of caspase, was analyzed to assess the role of XIAP in the regulation of caspase-3 in H/R-injured HCM supplemented with CM. Immunocytochemical staining of XIAP showed the localization of the protein in both the cytoplasmic and nuclear compartments of HCM. The XIAP staining seemed to be more concentrated and brighter in normoxic cells, while the staining was more diffused and less intense in H/R-injured cells (Figure 5). Quantitative immunofluorescence analysis revealed a high expression of XIAP in the normoxic cells (624.5 ± 69.6) as opposed to the lower expression of XIAP in all the H/R-injured cells treated with CM (between 325.9 ± 16.7 and 441.6 ± 11.4) (Figure 5). These results highlight the cellular impact of H/R injury on the downregulation of XIAP in HCM. Most importantly, no enhancement in XIAP expression following CM treatment suggests that CM derived from SkM-seeded EF–HAM scaffolds did not directly regulate the expression of XIAP in H/R-injured HCM, despite the antiapoptotic effect on caspase-3 as previously demonstrated.

### 2.6. CM Derived from SkM-Seeded HAM and EF–HAM Scaffolds Attenuated Mitochondrial Dysfunction in HCM Following H/R Injury

The impact of HAM CM and EF–HAM CM supplementation on the mitochondrial health of H/R-injured HCM was investigated by monitoring the mitochondrial membrane potential (ΔΨm) of cells using membrane-permeant JC-1 dye. As shown in Figure 6, cells in normoxic control and different CM treatment groups exhibited different shades of fluorescence depending on the amount of JC-1 monomer or aggregate formed in the cells. Cells supplemented with Non-CM and Plain CM produced a more green-yellowish fluorescence signal. Meanwhile, cells treated with CM derived from SkM-seeded HAM and EF–HAM scaffolds produced a more orange-brownish fluorescence signal.

The changes in ΔΨm between different treatment groups were quantitatively evaluated by determining the ratio of JC-1 monomer to aggregate indicated by the ratio of red to green fluorescence signal in labeled cells. A ratio higher than 1.0 indicates a healthy state of mitochondria, whereas a value less than 1.0 suggests that ΔΨm are being compromised. As displayed in Figure 6, the ratio of JC-1 red/green fluorescence for HAM CM and all EF–HAM CM treatment groups were all above 1.2, which was comparable to the normoxic control group (1.14 ± 0.01). Besides that, HCM treated with HAM CM and different EF–HAM CM also possessed significantly higher ΔΨm (*p* < 0.001) compared to the Non-CM control (0.92 ± 0.03) and Plain CM (0.91 ± 0.07) treatment group.

### 2.7. CM Derived from SkM-Seeded EF–HAM 5 Min Scaffolds Reduced the Formation of Reactive Oxygen Species in HCM Following H/R Injury

Dihydroethidium or DHE is a fluorescent probe used to directly measure the generation of intracellular superoxide in live cells. The levels of superoxide generated in H/R-injured HCM after CM treatment were measured through live imaging fluorescence microscopy. As shown in Figure 7, the DHE probe generated a bright red fluorescence signal when it was oxidized by the superoxide inside the cells. The levels of superoxide generated in HCM following H/R injury were significantly reduced when cells were treated with CM from SkM-seeded HAM and EF–HAM scaffolds, compared to the Non-CM control. The level of superoxide in normoxic cells (448.0 ± 0.9, *p* < 0.001) was also significantly lower compared to the Non-CM control (649.4 ± 44.9). Interestingly, only EF–HAM 5 min CM (378.0 ± 4.1, *p* < 0.0001) managed to significantly reduce the superoxide formation in injured cells when compared to Plain CM (540.4 ± 4.2). These results suggest that CM derived from SkM-seeded EF–HAM scaffolds might contain antioxidative factors that can reduce the generation of reactive oxygen species in HCM due to the H/R injury.

## 3. Discussion

The presented findings successfully described the rescue mechanism of primary human cardiomyocytes subjected to hypoxia–reoxygenation (H/R) injury, mediated by the paracrine secretion of skeletal muscle cells-seeded EF–HAM scaffolds. The H/R injury induces a series of adverse cellular events including mitochondrial damage, oxidative stress, and apoptosis, which collectively result in the death of cardiomyocytes. In this study, H/R injury was confirmed with the elevation of HIF-1α and impaired cell viability following 5 h of hypoxia and 6 h of reoxygenation. The CM derived from SkM-seeded EF–HAM scaffolds positively impacts hypoxia-/reoxygenation-induced changes in caspase-3 expression, mitochondrial membrane potential and reactive oxygen species generation, but not in XIAP expression.

The HIF-1α peaked after 5 h of hypoxia before it declined and increased again at 24 h. The 5 h hypoxic exposure time was chosen because a longer period of hypoxia (24 h) may cause irreversible damage to the cells, thus masking the real therapeutic benefits of CM on injured HCM. According to Hafez et al. (2018), HIF-1α was detected in the cytosol and nucleus of HCM after 3 h of hypoxia, but then stabilized and translocated within the nucleus after prolonged hypoxic exposure [6]. However, the expression of HIF-1α was not detected in normoxic cells [6], unlike in this study where baseline HIF-1α expression can still be detected in HCM despite not being subjected to hypoxia (0 h hypoxia). This circumstance could happen when healthy cardiomyocytes become overconfluent causing local hypoxia due to the high oxygen consumption rate.

After hypoxia, reoxygenation decreased cell viability in a time-dependent manner. A 6 h reoxygenation period was selected because it reduced the cell viability to approximately 50% when compared to normoxic control. A similar study was found to produce similar results whereby the cell viability of primary human cardiomyocytes decreased by 50% after 6 h of reoxygenation [6]. Nevertheless, optimal hypoxic/reoxygenation duration will always differ from study to study depending on the sensitivity of the cells to the insults. Hence, a series of optimizations on the exposure time is crucial before establishing the in vitro hypoxic-reoxygenation model.

When CM from SkM-loaded HAM and EF–HAM scaffolds were supplemented to the H/R-injured HCM, the CM exerted a pro-survival effect on HCM. On the other hand, our results also showed that supplementation of CM derived from SkM-seeded EF–HAM scaffolds on healthy HCM significantly increased the cell proliferation rate compared to the Non-CM control. These results suggest that CM derived from SkM-seeded EF–HAM scaffolds might contain important paracrine factors that could exert pro-survival and proliferative effects on healthy or injured HCM. The paracrine mechanism likely plays a significant role in the observed effects, where secreted factors, such as vascular endothelial growth factor (VEGF) [17], fibroblast growth factor (FGF) [18], and anti-inflammatory cytokines such as IL-10 [19], can enhance angiogenesis, reduce inflammation, and improve cardiomyocyte survival.

Additionally, the paracrine release of protective molecules like hepatocyte growth factor (HGF) and insulin-like growth factor 1 (IGF-1) may prevent apoptosis by activating pro-survival signaling pathways [20]. Moreover, the secretion of matrix metalloproteinases (MMPs) could support extracellular matrix remodeling, while the release of connexin proteins may enhance electrical coupling between cardiomyocytes [21]. Collectively, these paracrine interactions contribute to a microenvironment that supports the survival, function, and integration of primary human cardiomyocytes, promoting recovery after ischemic injury.

A previous study by Kadir et al. (2021) reported the benefit of aligned fiber orientation in influencing the production of the mesenchymal stem cell (MSC) secretome, which could promote cell proliferation and augment MSC differentiation [22]. Unfortunately, not much research has been performed on how HAM affects seeded cells’ ability to secrete pro-survival factors. However, a prior study by Faridvand et al. (2019) demonstrated that HAM extracts contain bioactive molecules that could confer protective effects against hypoxia-induced cell damage through the induction of heme oxygenase-1 (HO-1) and Nuclear factor erythroid 2-related factor 2 (Nrf2) [23]. To validate our theory, more research is necessary since the paracrine components in EF–HAM CM that give pro-survival and antiapoptotic effects on HCM have not been examined.

The H/R injury increased caspase-3 activity in injured cells compared to normoxic cells. Treatment of CM from SkM-seeded EF–HAM 5 min and EF–HAM 7 min scaffolds significantly reduced caspase-3 expression and the number of apoptotic cells. Caspase-3 induced apoptosis activation by reperfusion injury has been shown in mouse models. The treatment that salvaged the injured tissue, evidenced by restoration of cardiac output, also resulted in the reduction of caspase-3 [24]. Accordingly, this result indicates the expression of caspase-3 as a good marker of apoptotic cells and reperfusion injury. In this study, EF–HAM 5 min CM and EF–HAM 7 min CM effectively reduced caspase-3 expression in H/R-injured cells comparable to normoxic levels. Considering caspase-3 induces apoptosis via the proteolyzation of DNA repair proteins, cytoskeletal proteins and the inhibitor of Caspase-activated DNase (ICAD) [25], the inhibition of Caspase-3 via EF–HAM-derived CM implies its potential to attenuate cardiomyocyte apoptosis [26,27]. While the mechanism for CM-mediated antiapoptotic effect on injured HCM is not known, a previous study has demonstrated that myoblast-conditioned media containing HO-1, a heme-degrading enzyme, could improve muscle regeneration after ischemia by enhancing angiogenesis and reducing apoptosis [28].

Following H/R injury, XIAP expression in injured cardiomyocytes was decreased compared to the pre-injury control. Results from this study showed that the treatment of different CM did not influence the expression of XIAP. The effectiveness of directly targeting XIAP expression to inhibit cardiomyocyte apoptosis has been established in a previous study using microRNA [29]. Hypothetically, the EF–HAM CM is expected to inhibit apoptosis in HCM via the increased expression of XIAP in this study. Thus, the result may suggest that the cardiac rescue by EF–HAM CM after the H/R injury was not mediated by XIAP. Hence, future studies that investigate the interaction between XIAP and its target caspases are warranted to confirm this finding.

Following H/R injury, the JC-1 ratio decreased compared to the pre-injury control. Treatment of CM derived from SkM-seeded HAM and EF–HAM scaffolds restored the mitochondrial membrane potential (ΔΨm) of injured HCM to the pre-injury level. Mitochondria play key roles in the activation of apoptosis in cardiomyocytes since the activation signals of caspases are initiated in the mitochondria via the intrinsic pathway [30]. The mechanism underlying the CM-mediated stabilization of ΔΨm is unknown. Interestingly, in a study performed by Xiong et al. (2012), the implantation of a fibrin patch seeded with human embryonic stem cell-derived vascular cells (hESC–VCs) resulted in the improvements of ΔΨm, myocardial bioenergetics, and reduced apoptosis in the infarcted heart. It has also been shown that the internalization of released fragments from bioenergetic active material can produce metabolic intermediates, subsequently elevating ΔΨm, and accelerating tissue regeneration [31].

Following H/R injury, DHE was increased compared to the normoxic control. The DHE-positive cells indicated that the ischemic/reperfusion injury induced higher production of reactive oxygen species (ROS) [32]. Treatment of CM derived from SkM-seeded HAM and EF–HAM scaffolds reduced the intracellular ROS to the pre-injury level. In terms of bioactive molecules secreted by the cells such as EF–HAM derived CM, antioxidative properties could be exerted via the activation of the antioxidative enzyme such as superoxide dismutase, which is reported to diminish in the event of MI [33].

Additionally, previous studies have demonstrated that HAM contains bioactive molecules such as superoxide dismutase (SOD), catalase (CAT), and glutathione peroxidase (GPx), which contribute to the reduction of oxidative stress in injured tissues [23,33]. Mokhtari et al. (2020) showed that HAM-conditioned medium could significantly decrease myocardial injury markers, like cTn-I and MDA, while increasing antioxidant enzyme activity, thereby mitigating oxidative stress in ischemic myocardial tissues [34]. In addition, Faridvand et al. (2020) demonstrated that amnion membrane proteins (AMPs) could enhance mitochondrial membrane potential and antioxidative status in H9c2 cardiomyocytes, reducing the levels of oxidative damage markers such as ROS and MDA [35]. Moreover, Akseh et al. (2021) highlighted the potential anti-inflammatory effects of AMPs through the suppression of the TLR4/NF-κB pathway, which correlates with a reduction in pro-apoptotic signaling and oxidative stress [36]. Taken together, these studies indicate that the antioxidative capacity of HAM, particularly its ability to upregulate key antioxidant enzymes like SOD and CAT, plays a crucial role in alleviating oxidative stress and promoting myocardial survival after ischemia/reperfusion injury. Thus, the results suggest that the EF–HAM scaffold can reduce ROS production and attenuate oxidative stress in ischemic cardiomyocytes via paracrine secretion.

In conclusion, the EF–HAM scaffold exhibits significant potential for ischemic cardiac repair, primarily due to its cardioprotective and antiapoptotic properties. This innovative cardiac patch combines human amniotic membrane and electrospun PLGA fibers, offering mechanical support and mimicking the structure of cardiac tissue [14,15]. Notably, results from previous and current studies demonstrate that EF–HAM scaffolds are cardioprotective, promote cell viability and survival, enhance mitochondrial health, protect against myocardial apoptosis and alleviate cellular oxidative stress in injured cardiomyocytes, and promote angiogenesis through paracrine signaling, as shown in a prior in vitro study [16].

The EF–HAM patch, when seeded with a patient’s healthy skeletal muscle cells, can be used in conjunction with Coronary Artery Bypass Grafting (CABG) surgery to support and regenerate damaged heart tissue. The scaffold’s ability to integrate with epicardial tissue and its secretion of therapeutic factors make it a promising solution for improving outcomes in patients with myocardial infarction. Future research should focus on optimizing the integration of skeletal muscle cells with the EF–HAM patch, refining cell-seeding techniques, and evaluating long-term cardiac outcomes post-CABG in clinical trials. Such studies could include parameters like improved myocardial perfusion, electrical synchronization, and overall patient survival rates to substantiate the broader therapeutic potential of this strategy.

Finally, the current study reliance on fluorescent staining for quantifying markers such as HIF, caspases, ROS, and JC-1 limits the robustness of the findings. Future research should include complementary techniques like ELISA, PCR, or Western blotting to provide a more precise quantification of these markers and enhance the reliability of the results. Additionally, further studies are needed to fully elucidate the paracrine factors responsible for the observed cardioprotective effects of EF–HAM CM.

## 4. Materials and Methods

### 4.1. Ethical Consideration

This study was approved by the Universiti Kebangsaan Malaysia Research Ethics Committee (UKMREC) with the approval code of UKM 1.5.3.5/244/02-01-02-SF1284 and UKM PPI/111/8/JEP-2017-411.

### 4.2. Fabrication of EF–HAM Scaffolds

The fabrication of EF–HAM scaffolds involved the collection, processing, and decellularization of the human amniotic membrane (HAM), followed by the electrospinning process on the decellularized HAM as described previously [14,16]. The HAM was collected from healthy, consented mothers undergoing caesarean sections, ensuring no medical or obstetric complications. The redundant tissue was placed in a sterile 60 mL specimen container with DPBS + 1% AA solution and transported to the laboratory. The HAM was separated from the chorion layer using blunt dissection and then extensively rinsed in DPBS to remove debris, mucus, and blood stains. The tissue was soaked in 0.05% NaClO solution with moderate shaking for 1 h and subsequently washed in DPBS for 3 × 15 min with moderate shaking to remove excess NaClO. The cleaned HAM was transferred to a 50 mL Falcon tube containing 30 mL DPBS with 1% AA solution and frozen at −80 °C.

Prior to the decellularization steps, the frozen HAM was thawed at room temperature and cut into 3 cm x 3 cm pieces. A 18.75 µg/mL thermolysin solution (TL) was preheated at 60 °C for 10 min. The HAM pieces were incubated in the preheated TL solution at 37 °C for 10 min, followed by pulse vortexing 60 times. The TL was removed by washing HAM in DPBS for 3 × 15 min with moderate shaking (150 rpm). The HAM was then soaked in 0.25 M NaOH for 1 min, followed by pulse vortexing 30 times. Excess NaOH was removed by washing the HAM in DPBS for 3 × 15 min with moderate shaking (150 rpm). The decellularized HAM pieces were air-dried under a biosafety cabinet for at least 1.5 h and sterilized using gamma radiation at 15 kGy. The sterilized HAM pieces were stored at room temperature in a dry storage area.

For the fabrication of EF–HAM scaffolds via electrospinning, a solvent mixture of DCM and DMF in a 7:3 volume ratio was prepared. The PLGA 50:50 polymer was dissolved in this solvent mixture to make a 20% (*w*/*v*) solution, stirred at 230 rpm for 20 h at room temperature. The PLGA solution was loaded into a 1 mL syringe with a 21 G blunt needle and electrospun onto decellularized HAM attached to aluminum foil on a rotating collector at an applied voltage of 7.5 kV, polymer flow rate of 0.3 mL/h, and deposition distance of 15 cm. The rotating collector was run at 1000 rpm to create aligned PLGA fibers on HAM. Fiber thickness variations were achieved by adjusting the deposition times to 3 min, 5 min, and 7 min.

The resulting EF–HAM scaffolds were air-dried under a biosafety cabinet for 1.5 h to allow for complete solvent evaporation, followed by UV irradiation for 40 min. The scaffolds were stored at 4 °C for no more than one week. Before experimental assays, all HAM and EF–HAM scaffolds were hydrated and carefully peeled off the aluminum foils. The scaffolds were wrapped around non-stick Teflon rings with an inner diameter of 16 mm, fitting into a 12-well tissue culture plate, with the fiber-coated sides of EF–HAM facing upwards. This assembly prevented the scaffolds from floating in the cell culture media and the PLGA fibers from detaching from HAM.

### 4.3. Skeletal Muscle Cells Isolation

Skeletal muscle cells (SkMs) were isolated from redundant muscle tissues obtained during surgeries such as wound debridements, amputations, or traumatic injuries at Hospital Canselor Tuanku Muhriz UKM (HCTM). Patients received information sheets and provided informed consent before tissue collection. There were no specific donor criteria if the tissue was redundant and consented. Tissue processing used the trypsin–EDTA method. In brief, skeletal muscle tissue was rinsed in DPBS, cleared of debris, minced, and digested with 0.25% trypsin–EDTA at 37 °C for 10 min. Isolated cells were then separated from undigested tissue by centrifugation, resuspended in a mix of Ham’s F10 and DMEM + 20% FBS, and cultured at 37 °C with 5% CO_2_. Digestion was repeated twice with undigested tissue. From passage 1 onwards, cells were cultured in F10 + 10% FBS on laminin-coated surfaces to enrich myoblasts, with medium changes every 48 h [14,16].

### 4.4. Conditioned Medium Collection

Skeletal myoblast-enriched SkMs were harvested at passage 3 and seeded on plain culture surfaces, HAM, and EF–HAM scaffolds at a seeding density of 20,000 cells/cm². They were grown for 5 days in an F10:DMEM medium at 37 °C in an atmosphere with 5% CO_2_. The waste medium was then removed, and the cells were briefly washed twice with serum-free F10:DMEM before replacing it with fresh serum-free F10:DMEM at 300 µL/cm^2^ of scaffold (total volume of 600 µL per scaffold). After 72 h of media conditioning, the conditioned media (CM) were collected from the scaffolds and centrifuged at 5000 rpm for 5 min to remove cell debris. The collected CM were labeled as Plain CM, HAM CM, EF–HAM 3 min CM, EF–HAM 5 min CM, and EF–HAM 7 min CM. A Non-CM was included as a negative control for all subsequent experiments, as it was not conditioned with either SkM or scaffold and thus contained no paracrine factors. All CM were stored at −80 °C until further use to prevent protein degradation.

### 4.5. Cell Viability Assay

The Cell Counting Kit-8 (CCK-8; Dojindo, Rockville, MD, USA) assay was performed to determine the viability of healthy primary human cardiomyocytes (HCMs) exposed to different conditioned media (CM) from skeletal muscle cells (SkMs)-seeded human amniotic membrane (HAM) and electrospun fiber-coated human amniotic membrane (EF–HAM) scaffolds. The HCMs were seeded into a 24-well plate at a seeding density of 6000 cells/cm^2^ and cultured in 260 µL Dulbecco’s Modified Eagle’s Medium (DMEM) + 10% Fetal Bovine Serum (FBS) medium supplemented with 130 µL CM (33% *v*/*v*) at 37 °C and 5% CO_2_. On Day 2 and Day 5 of the culture, the viability of cells was assayed through incubation with 10% (*v*/*v*) CCK-8 reagent for 3 h at 37 °C and 5% CO_2_ in the dark. Absorbance reading was performed at 450 nm by spectrophotometer.

### 4.6. In Vitro Model of Cardiac Ischemia-Reperfusion Injury

Primary HCM isolated from the ventricles of adult heart was purchased from PromoCell (Heidelberg, Germany) and supplied in a cryopreserved ampule containing ≥ 500,000 cells. HCM were seeded into a 24-well plate (Greiner Bio-One, Frickenhausen, Germany) at a seeding density of 6000 cells/cm^2^ and grown for 48 h in DMEM + 10% FBS in a humidified incubator (37 °C, 5% CO_2_). Then, the HCMs were subjected to hypoxic culture conditions followed by reoxygenation to induce cell damage and obtained ischemia-reperfusion injury in vitro.

#### 4.6.1. Hypoxic Induction Phase and HIF-1α Detection

The culture medium for HCM was changed from serum-supplemented DMEM to serum-free, glucose-free and phenol-free DMEM (Thermo Fisher Scientific, Waltham, MA, USA), which had been preconditioned under hypoxic conditions for 24 h to completely remove dissolved oxygen. The HCM was later placed into an oxygen control incubator (Galaxy 170 R, New Brunswick, NJ, USA) and incubated under hypoxic conditions (1% O_2_, 5% CO_2_ and 94% N_2_) for an experimental duration of 0, 3, 5, 10, 19, and 24 h.

The optimal exposure time for hypoxia was analyzed by performing immunocytochemical staining which was used to measure the expression of HIF-1α, a transcriptional regulator of the adaptive cellular response to hypoxia. The HCMs were stained with rabbit monoclonal anti-human HIF-1α antibody (1:250 dilution; Abcam, UK), followed by immunolabeling with goat anti-rabbit IgG (H+L) secondary antibody–Alexa Fluor^®^ 488 (1:300 dilution), and counterstaining with DAPI and Phalloidin. Images of cells stained with HIF-1α were captured under a fluorescence microscope and later analyzed to measure the mean fluorescence intensity (MFI) of the signals.

The MFI was measured using the ImageJ software to obtain the region of interest (ROI). The ROI with positive signals was selected and outlined, and the MFI of the ROI was later measured. The shortest hypoxic duration that could induce the highest expression of HIF-1α in HCM was chosen as the optimal exposure time for hypoxia.

#### 4.6.2. Reoxygenation Phase

After being exposed to optimal hypoxic duration, HCMs were reoxygenated and incubated in DMEM + 10% FBS under normoxic conditions (21% O_2_ and 5% CO_2_) for 3, 6, 9, 16, and 24 h. The cytotoxic effect of post-hypoxic reoxygenation on HCM viability was determined through CCK-8 assay. At the end of the reoxygenation period, 10% (*v*/*v*) CCK-8 reagent was directly added to the culture medium, and samples were incubated for 3 h at 37 °C and 5% CO_2_ in the dark. The CCK-8 absorbance for each treatment group was measured at 450 nm by spectrophotometer and the values were normalized to the cell viability under normoxic conditions, which was set at 100%.

### 4.7. Hypoxia–Reoxygenation (H/R) Injury Rescue in HCM

The ischemic rescue potential of CM derived from SkM-seeded HAM and EF–HAM scaffolds on HCM with H/R-induced injury was studied by using the established in vitro hypoxia-reoxygenation (H/R) injury model. Accordingly, the HCM was exposed to 5 h of hypoxia, followed by a 6 h reoxygenation period, during which 50% (*v*/*v*) CM was supplemented to the cells. Subsequently, the outcomes from this study included cell survival analysis, caspase-3 and XIAP expression profile, mitochondrial membrane potential analysis and measurement of reactive oxygen species (ROS).

#### 4.7.1. Cell Survival Analysis

The pro-survival effect of CM on HCM subjected to H/R-induced injury was determined through the measurement of cell viability by CCK-8. After the reoxygenation phase, the viability of HCM supplemented with CM was assayed through the incubation of cells with 10% (*v*/*v*) CCK-8 reagent for 3 h at 37 °C and 5% CO_2_. Absorbance reading was performed at 450 nm by spectrophotometer. Cells grown under normoxic conditions served as the reference for healthy uninjured cells.

#### 4.7.2. Caspase-3 Detection

The potential of CM to rescue H/R-injured HCM from apoptosis was studied by probing cells with BioTracker NucView^®^ 488 Green Caspase-3 dye (Sigma-Aldrich, St. Louis, MA, USA) to detect apoptotic cells. At the end of the reoxygenation phase, the HCM culture medium was replaced with Hank’s 1× Balanced Salt Solution (HBSS) containing 5 µM Caspase-3 dye and 1 µg/mL Hoechst dye, before samples were incubated at room temperature for 30 min in the dark. Stained samples were viewed under a fluorescence microscope and the percentage of cells stained positive with caspase-3 was calculated.

#### 4.7.3. XIAP Expression

After the reoxygenation phase, HCMs were stained with XIAP antibody to study its correlation with caspase-3 expression. The HCMs were incubated with mouse monoclonal anti-human XIAP (A-7) antibody (1:50 dilution; Santa Cruz Biotechnology, Dallas, TX, USA), followed by immunolabeling by goat anti-mouse Alexa Fluor^®^ 594 secondary antibody (1:300 dilution) and counterstaining with DAPI. Images of immunolabeled cells were captured under a fluorescence microscope and later analyzed to measure the mean fluorescence intensity (MFI) of XIAP for each treatment group

#### 4.7.4. Mitochondrial Membrane Potential Analysis

After the reoxygenation phase, JC-1 dye (MedChemExpress, Princeton, NJ, USA), at a stock concentration of 200 µM, was added directly to the CM-containing medium to make the final concentration at 2 μM, prior to incubation with cells at 37 °C, 5% CO_2_, for 30 min. After rinsing with Dulbecco’s Phosphate Buffered Saline (DPBS), JC-1 monomers and aggregates in labeled cells were detected at the excitation wavelength of 485 nm and 535 nm, respectively, under a fluorescence microscope in HBSS. Fluorescence images were later analyzed using ImageJ to measure the mean fluorescence intensity (MFI) of JC-1 and calculate the ratio of red/green fluorescence for each treatment group.

#### 4.7.5. Measurement of Reactive Oxygen Species

After the reoxygenation phase, cells were rinsed twice in DPBS and later incubated with HBSS containing 10 µM DHE (MedChemExpress, Princeton, NJ, USA) at 37 °C, 5% CO_2_, for 15 min. After rinsing in DPBS, stained cells were viewed under a fluorescence microscope in a fresh DMEM culture medium. Fluorescence images were later analyzed using ImageJ to measure the mean fluorescence intensity (MFI) of DHE.

### 4.8. Mean Fluorescence Intensity (MFI) Analysis

The mean fluorescence intensity (MFI) of biological indicators from fluorescent images were measured using an image processing and analysis software called the ImageJ version 1.53p (National Institutes of Health, Bethesda, MD, USA). All images were captured under the same magnification to standardize the field of view across different experimental groups. Prior to image analysis, the ND2 file was imported into ImageJ using Bio-Formats plugin and split into separate fluorescence channels. Next, background subtraction was performed on the image using the rolling ball radius plugin set at 50.0 pixels. Then, automatic image thresholding was run on the image using Otsu algorithm to identify and subsequently exclude background signals, creating a specific region of interest (ROI) on the image. The ROI with positive signals was then selected and outlined, and the net MFI within the ROI was later measured. The total MFI was then normalized to the number of cells within the ROI to yield the average MFI per cell.

### 4.9. Statistical Analysis

All data were presented as mean ± standard error of the mean. Each assay or analysis had been performed on three independent biological samples and run-on triplicate unless stated otherwise in the methodology sections. Statistical analysis was performed using GraphPad Prism 7.0 (Graph Pad Software, Boston, MA, USA). One-way analysis of variance (ANOVA) was used to compare the results of multiple groups with one independent variable, while two-way ANOVA was used for datasets with two independent variables. A *p*-value that was less than 0.05 was considered statistically significant.

## Figures and Tables

**Figure 1 ijms-25-11697-f001:**
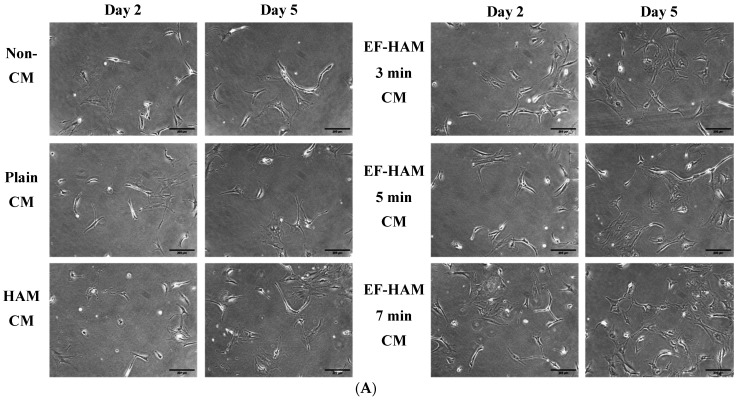
The effect of CM derived from SkM-seeded HAM and EF–HAM scaffolds on the viability and proliferation of healthy primary human cardiomyocytes. (**A**) Representative images of healthy primary human cardiomyocytes (HCMs) supplemented with 33% (*v*/*v*) CM derived from SkM-seeded HAM and EF–HAM scaffolds at day 2 and day 5 of culture under normoxic condition (10× magnification; 200 µm scale bar; *n* = 3). (**B**) The bar graph represents the cell viability results between day 2 and day 5 culture. The HCM viability was measured by incubating cells with 10% (*v*/*v*) CCK-8 reagent and absorbance reading at 450 nm. The values are represented as mean ± SEM (*n* = 3; * *p* < 0.05; one-way ANOVA). (**C**) The proliferation rate of HCM was enhanced in the presence of different EF–HAM CM compared with the Non-CM control. The values are represented as mean ± SEM (*n* = 3; * *p* < 0.05, compared with the Non-CM control; one-way ANOVA).

**Figure 2 ijms-25-11697-f002:**
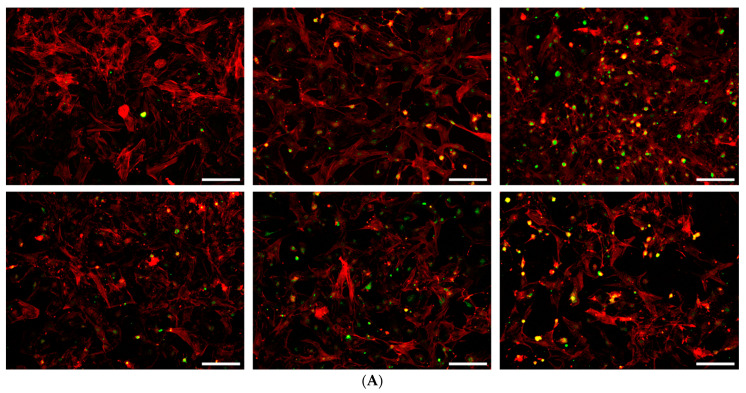
The establishment of in vitro cardiac ischemia–reperfusion injury model. (**A**) Immunofluorescence images of HCM stained with HIF-1α (green) and phalloidin (red) after being exposed to hypoxia (1% O_2_, 5% CO_2_ and 94% N_2_) for 0, 3, 5, 10, 19, and 24 h (10× magnification; 200 µm scale bar; *n* = 3). (**B**) Mean fluorescence intensity (MFI) of HIF-1α in HCM after different periods of hypoxia. The optimal exposure time for hypoxia was 5 h since it induced the highest expression of HIF-1α in hypoxic HCM. The values are represented as mean ± SEM (*n* = 3). (**C**) Post-hypoxic reoxygenation proportionally reduced HCM viability in a time dependent manner. The HCM was exposed to hypoxia for 5 h and subsequently reoxygenated for different time periods to determine optimal reoxygenation time at approximately 50% cell viability (indicated by the horizontal dotted red line). The values are represented as mean ± SEM (*n* = 3; **** *p* < 0.0001, compared with Normoxic control; one-way ANOVA).

**Figure 3 ijms-25-11697-f003:**
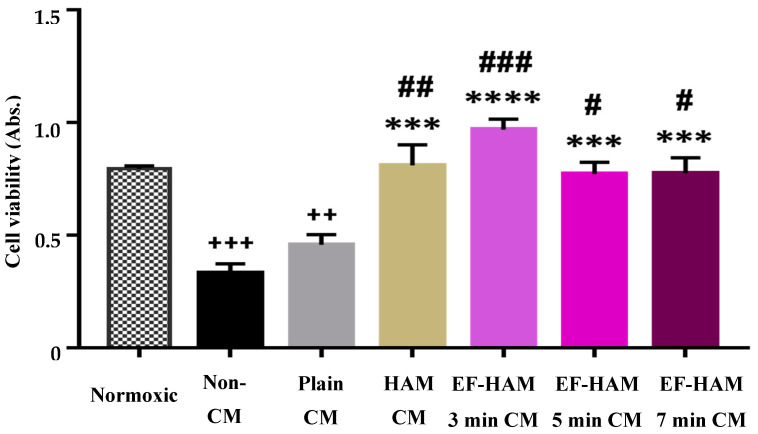
The pro-survival effect of CM derived from SkM-seeded HAM and EF–HAM scaffolds on the viability of HCM following H/R-induced injury. The values are represented as mean ± SEM (*n* = 3; ++ *p* < 0.01 or +++ *p* < 0.001, compared with Normoxic; *** *p* < 0.001 or **** *p* < 0.0001, compared with Non-CM control; # *p* < 0.05, ## *p* < 0.01 or ### *p* < 0.001, compared with Plain CM; one-way ANOVA).

**Figure 4 ijms-25-11697-f004:**
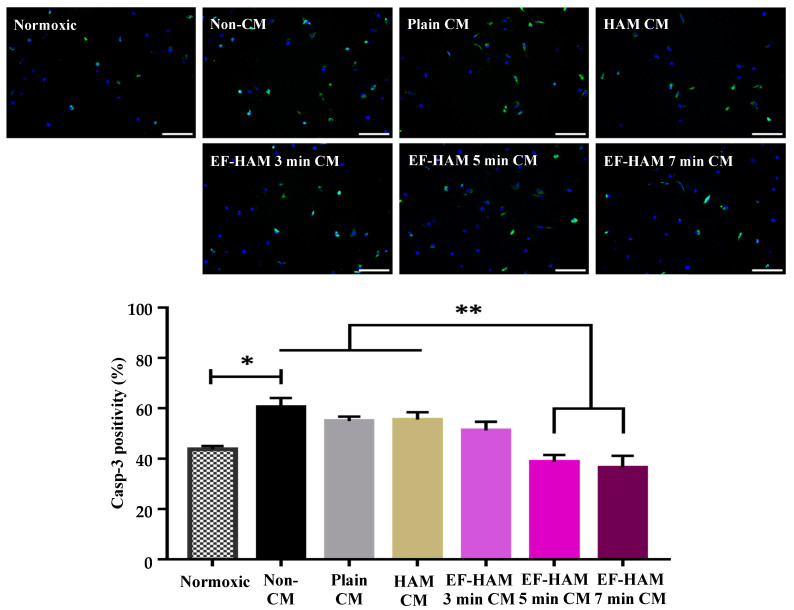
The effects of CM on HCM subjected to H/R-induced injury according to the caspase-3 expression profile. Immunofluorescence images of HCM stained with 5 µM caspase-3 probe (green) and 1 µg/mL Hoescht dye (blue) after being treated with different CM following H/R-induced injury (10× magnification; 200 µm scale bar; *n* = 3). The bar graph represents the percentage of caspase-3 positive cells after HCM subjected to H/R-induced injury were rescued with CM treatment. The values are represented as mean ± SEM (*n* = 3; * *p* < 0.05 or ** *p* < 0.01; one-way ANOVA).

**Figure 5 ijms-25-11697-f005:**
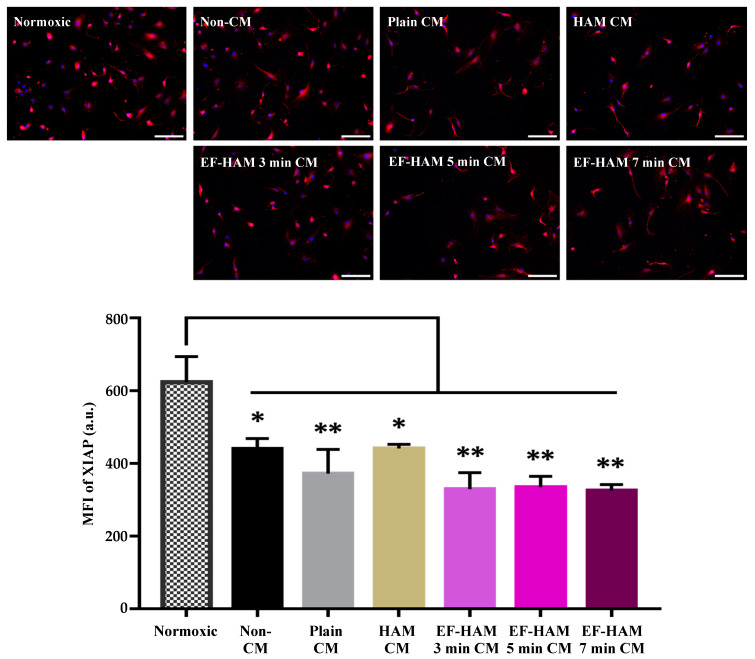
The effects of CM on HCM subjected to H/R-induced injury according to the XIAP expression profile. Immunofluorescence images of HCM stained with XIAP antibody (red) and DAPI (blue) after being treated with different CM following H/R-induced injury (10× magnification; 200 µm scale bar; *n* = 3). The bar graph represents the MFI measurement for XIAP in H/R-injured HCM after ischemic rescue with CM. Treatment with different EF–HAM CM did not upregulate the expression of XIAP injured HCM. The values are represented as mean ± SEM (*n* = 3; * *p* < 0.05 or ** *p* < 0.01, compared with Normoxic control; one-way ANOVA).

**Figure 6 ijms-25-11697-f006:**
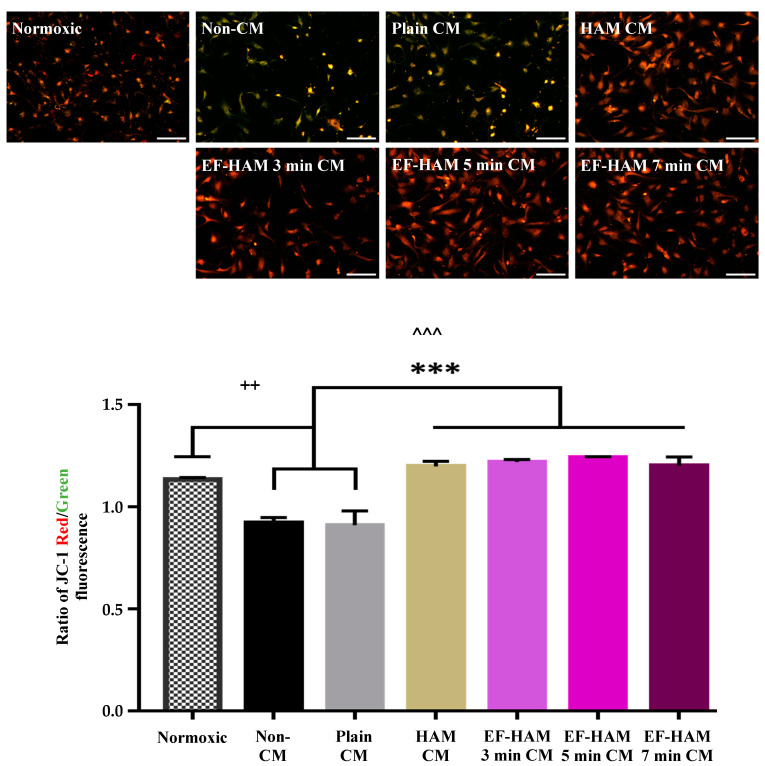
The effects of CM on HCM subjected to H/R-induced injury according to the mitochondrial membrane potential analysis. Immunofluorescence images of HCM stained with 2 µM JC-1 probe (red/green) after being treated with different CM following H/R-induced injury (10× magnification; 200 µm scale bar; *n* = 3). The bar graph represents the ratio of JC-1 red/green fluorescence in H/R-injured HCM after ischemic rescue with CM. Treatment with CM derived from SkM-seeded HAM and EF–HAM scaffolds enhanced the mitochondrial membrane potential in injured HCM. The values are represented as mean ± SEM (*n* = 3; ++ *p* < 0.01, compared with Normoxic; ^^^ *p* < 0.001, compared with Non-CM control; and *** *p* < 0.001, compared with Plain CM group; one-way ANOVA).

**Figure 7 ijms-25-11697-f007:**
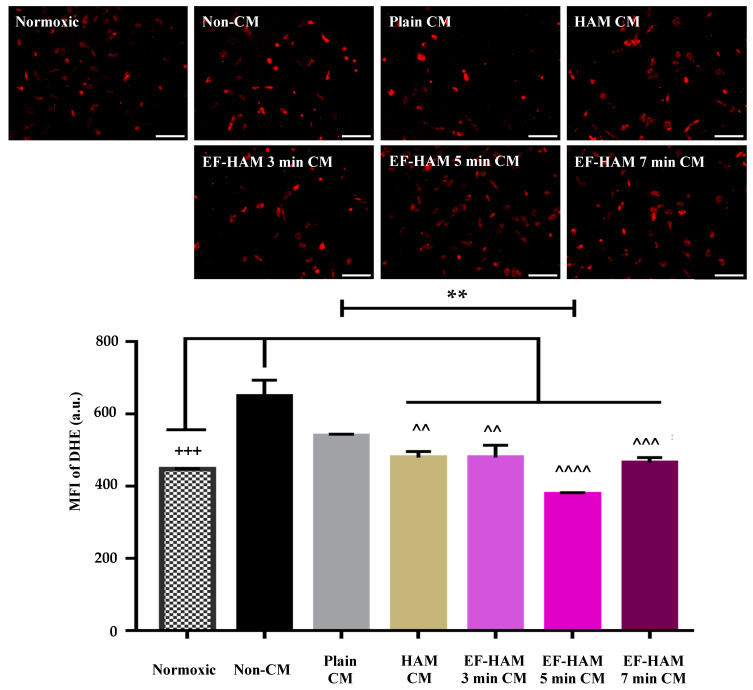
The effects of CM on HCM subjected to H/R-induced injury according to the measurement of reactive oxygen species (ROS). Immunofluorescence images of HCM stained with 10 µM DHE dye (red) after being treated with different CM following H/R-induced injury (10× magnification; 200 µm scale bar; *n* = 3). The bar graph represents the MFI measurement for DHE in H/R-injured HCM after ischemic rescue with CM. Treatment with CM derived from SkM-seeded HAM and EF–HAM scaffolds reduced the oxidative stress in injured HCM. The values are represented as mean ± SEM (*n* = 3; +++ *p* < 0.001, compared with Normoxic; ^^ *p* < 0.01, ^^^ *p* < 0.001 or ^^^^ *p* < 0.0001, compared with Non-CM control; and ** *p* < 0.01, compared with Plain CM group; one-way ANOVA).

## Data Availability

The raw data supporting the conclusions of this article will be made available by the authors on request.

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
