# Peer review of "Ischemic Rescue Potential of Conditioned Medium Derived from Skeletal Muscle Cells-Seeded Electrospun Fiber-Coated Human Amniotic Membrane Scaffolds"

_ijms, 2024, doi:10.3390/ijms252111697_

Round 1
Reviewer 1 Report
Comments and Suggestions for Authors
Thank you for the opportunity to review manuscript titled “Ischemic Rescue Potential of Conditioned Medium Derived From Skeletal Muscle Cells-Seeded Electrospun Fiber-Coated Human Amniotic Membrane Scaffolds” The study presents interesting findings; however, I have several comments that I believe could enhance the clarity and impact of the work:
- The manuscript does not specify whether the cells used are established cell lines or primary cells derived from donors. Please clarify the source of the cells in the methods section.
- In Figure 3D, the symbols ^ and + are not defined in the figure legend.
- In Figure 2B, the purpose of the horizontal dotted line is unclear. Additionally, in Figure 2C, the horizontal red line also lacks explanation.
- Figure 3 contains nine separate figures, making it confusing. I recommend to split it into four separate figures, with each fluorescent image accompanied by its quantitative analysis
- The discussion of the amniotic membrane and its role as an antioxidant could be enhanced and better supported with additional references and evidence.
- A notable weakness of the study is its primary reliance on fluorescent staining for quantifying key markers such as HIF, caspases, ROS, and JC-1. The absence of complementary techniques like ELISA, PCR, or Western blotting limits the robustness of the findings. (SHOULD BE ADDED AS LIMITATION AT THE EDN OF THE STUDY)
- Future investigations regarding the use of skeletal muscle cells in conjunction with Coronary Artery Bypass Grafting (CABG) should be mentioned to highlight the broader implications of the research.
- 2023-2024 citations are missing
Author Response
Thank you very much for taking the time to review this manuscript. Please find detailed responses in the files attached below and the corresponding revisions/corrections highlighted/in track changes in the re-submitted files.

Reviewer 2 Report
Comments and Suggestions for Authors
Dear authors,
The manuscript is well written and structured, however some details need to be clarified before publication:
What are the primary revascularization techniques mentioned in the study, and what adverse effects are they known to cause?
What is the main objective of the present study involving primary human cardiomyocytes?
How was the in vitro cardiac ischemia-reperfusion injury model established in the study?
What types of analyses were conducted to evaluate the outcomes of the study?
How did the viability of human cardiomyocytes change when exposed to different conditioned media?
Which specific conditioned media showed the highest increase in cell viability, and what were the percentages?
What impact did the conditioned media derived from skeletal muscle-seeded EF-HAM scaffolds have on caspase-3 expression, mitochondrial membrane potential, and reactive oxygen species generation?
Did the conditioned media have any effect on XIAP expression in hypoxia/reperfusion-induced primary human cardiomyocytes?
What conclusions can be drawn about the anti-apoptotic and cardioregenerative effects of EF-HAM composite scaffolds based on the study’s findings?
How might the paracrine mechanism contribute to the observed effects of EF-HAM composite scaffolds on primary human cardiomyocytes?
Author Response

(The authors gave the same response as above.)

Round 2
Reviewer 1 Report
Comments and Suggestions for Authors
No further comments
Reviewer 2 Report
Comments and Suggestions for Authors
Dear Authors,
I recommend the publication of the manuscript in the present form, having in account that all the comments of the reviewers were answered.